# Quadratic Video Interpolation

**Xiangyu Xu**[*†]
Carnegie Mellon University
xuxiangyu2014@gmail.com

**Li Siyao**[*]
SenseTime Research
lisiyao1@sensetime.com

**Wenxiu Sun**
SenseTime Research
sunwenxiu@sensetime.com

**Qian Yin**
Beijing Normal University
yinqian@bnu.edu.cn

**Ming-Hsuan Yang**
University of California, Merced   Google
mhyang@ucmerced.edu

## Abstract

Video interpolation is an important problem in computer vision, which helps overcome the temporal limitation of camera sensors. Existing video interpolation methods usually assume uniform motion between consecutive frames and use linear models for interpolation, which cannot well approximate the complex motion in the real world. To address these issues, we propose a quadratic video interpolation method which exploits the acceleration information in videos. This method allows prediction with curvilinear trajectory and variable velocity, and generates more accurate interpolation results. For high-quality frame synthesis, we develop a flow reversal layer to estimate flow fields starting from the unknown target frame to the source frame. In addition, we present techniques for flow refinement. Extensive experiments demonstrate that our approach performs favorably against the existing linear models on a wide variety of video datasets.

## 1 Introduction

Video interpolation aims to synthesize intermediate frames between the original input images, which can temporally upsample low-frame rate videos to higher-frame rates. It is a fundamental problem in computer vision as it helps overcome the temporal limitations of camera sensors and can be used in numerous applications, such as motion deblurring [5, 35], video editing [25, 38], virtual reality [1], and medical imaging [11].

Most state-of-the-art video interpolation methods [2, 3, 9, 14, 17] explicitly or implicitly assume uniform motion between consecutive frames, where the objects move along a straight line at a constant speed. As such, these approaches usually adopt linear models for synthesizing intermediate frames. However, the motion in real scenarios can be complex and non-uniform, and the uniform assumption may not always hold in the input videos, which often leads to inaccurate interpolation results. Moreover, the existing models are mainly developed based on two consecutive frames for interpolation, and the higher-order motion information of the video (*e.g.,* acceleration) has not been well exploited. An effective frame interpolation algorithm should use additional input frames and estimate the higher-order information for more accurate motion prediction.

To this end, we propose a quadratic video interpolation method to exploit additional input frames to overcome the limitations of linear models. Specifically, we develop a data-driven model which integrates convolutional neural networks (CNNs) [13, 24] and quadratic models [15] for accurate motion estimation and image synthesis. The proposed algorithm is acceleration-aware, and thus

---

[*]Equal contributions.
[†]Corresponding author.

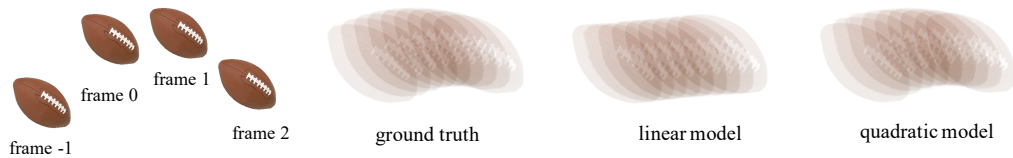

Figure 1: Exploiting the quadratic model for acceleration-aware video interpolation. The leftmost subfigure shows four consecutive frames from a video, describing the projectile motion of a football. The other three subfigures show the interpolated results between frame 0 and 1 by different algorithms. Note that we overlap these results for better visualizing the interpolation trajectories. Since the linear model [31] assumes uniform motion between the two frames, it does not approximate the movement in real world well. In contrast, our quadratic approach can exploit the acceleration information from the four neighboring frames and generate more accurate in-between video frames.

allows predictions with curvilinear trajectory and variable velocity. Although the ideas of our method are intuitive and sensible, this task is challenging as we need to estimate the flow field from the unknown target frame to the source frame (*i.e.,* backward flow) for image synthesis, which cannot be easily obtained with existing approaches. To address this issue, we propose a flow reversal layer to effectively convert forward flow to backward flow. In addition, we introduce new techniques for filtering the estimated flow maps. As shown in Figure 1, the proposed quadratic model can better approximate pixel motion in real world and thus obtain more accurate interpolation results.

The contributions of this work can be summarized as follows. First, we propose a quadratic interpolation algorithm for synthesizing accurate intermediate video frames. Our method exploits the acceleration information of the video, which can better model the nonlinear movements in the real world. Second, we develop a flow reversal layer to estimate the flow field from the target frame to the source frame, thereby facilitating high-quality frame synthesis. In addition, we present novel techniques for refining flow fields in the proposed method. We demonstrate that our method performs favorably against the state-of-the-art video interpolation methods on different video datasets. While we focus on quadratic functions in this work, the proposed framework for exploiting the acceleration information is general, and can be further extended to higher-order models.

## 2   Related Work

Most state-of-the-art approaches [2, 3, 4, 9, 14, 17, 19] for video interpolation explicitly or implicitly assume uniform motion between consecutive frames. As a typical example, Baker *et al.* [2] use optical flow and forward warping to linearly move pixels to the intermediate frames. Liu *et al.* [14] develop a CNN model to directly learn the uniform motion for interpolating the middle frame. Similarly, Jiang *et al.* [9] explicitly assume uniform motion with flow estimation networks, which enables a multi-frame interpolation model.

On the other hand, Meyer *et al.* [17] develop a phase-based method to combine the phase information across different levels of a multi-scale pyramid, where the phase is modeled as a linear function of time with the implicit uniform motion assumption. Since the above linear approaches do not exploit higher-order information in videos, the interpolation results are less accurate.

Kernel-based algorithms [20, 21, 33] have also been proposed for frame interpolation. While these methods are not constrained by the uniform motion model, existing schemes do not handle nonlinear motion in complex scenarios well as only the visual information of two consecutive frames is used for interpolation.

Closely related to our work is the method by McAllister and Roulier [15] which uses quadratic splines for data interpolation to preserve the convexity of the input. However, this method can only be applied to low-dimensional data, while we solve the problem of video interpolation which is in much higher dimensions.

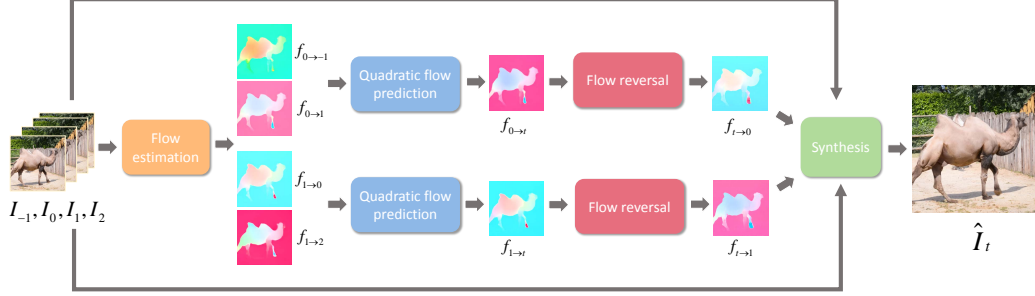

Figure 2: Overview of the quadratic video interpolation algorithm. We first use the off-the-shelf model to estimate flow fields for the input frames. Then we introduce quadratic flow prediction and flow reversal layers to estimate $\boldsymbol{f}_{t\to 0}$ and $\boldsymbol{f}_{t\to 1}$. We describe the estimation process of $\boldsymbol{f}_{t\to 0}$ in details in this paper, and $\boldsymbol{f}_{t\to 1}$ can be computed similarly. Finally, we synthesize the in-between frame by warping and fusing the input frames with $\boldsymbol{f}_{t\to 0}$ and $\boldsymbol{f}_{t\to 1}$.

## 3 Proposed Algorithm

To synthesize an intermediate frame $\hat{I}_t$ where $t \in (0, 1)$, existing algorithms [9, 14, 21] usually assume uniform motion between the two consecutive frames $I_0, I_1$, and adopt linear models for interpolation. However, this assumption cannot approximate the complex motion in real world well and often leads to inaccurately interpolated results. To solve this problem, we propose a quadratic interpolation method for predicting more accurate intermediate frames. The proposed method is acceleration-aware, and thus can better approximate real-world scene motion.

An overview of our quadratic interpolation algorithm is shown in Figure 2, where we synthesize the frame $\hat{I}_t$ by fusing pixels warped from $I_0$ and $I_1$. We use $I_{-1}$, $I_0$, and $I_1$ to warp pixels from $I_0$ and describe this part in details in the following sections, and the warping from the other side (*i.e.,* $I_1$) can be performed similarly by using $I_0$, $I_1$, and $I_2$. Specifically, we first compute optical flow $\boldsymbol{f}_{0\to 1}$ and $\boldsymbol{f}_{0\to -1}$ with the state-of-the-art flow estimation network PWC-Net [31]. We then predict the intermediate flow map $\boldsymbol{f}_{0\to t}$ using $\boldsymbol{f}_{0\to 1}$ and $\boldsymbol{f}_{0\to -1}$ in Section 3.1. In Section 3.2, we propose a new method to estimate the backward flow $\boldsymbol{f}_{t\to 0}$ by reversing the forward flow $\boldsymbol{f}_{0\to t}$. Finally, we synthesize the interpolated results with the backward flow in Section 3.3.

### 3.1 Quadratic flow prediction

To interpolate frame $\hat{I}_t$, we first consider the motion model of a pixel from $I_0$:

$$\boldsymbol{f}_{0\to t} = \int_0^t \left[ \boldsymbol{v}_0 + \int_0^\kappa \boldsymbol{a}_\tau d\tau \right] d\kappa, \tag{1}$$

where $\boldsymbol{f}_{0\to t}$ denotes the displacement of the pixel from frame 0 to $t$, $\boldsymbol{v}_0$ is the velocity at frame 0, and $\boldsymbol{a}_\tau$ represents the acceleration at frame $\tau$.

Existing models [9, 14, 21] usually explicitly or implicitly assume uniform motion and set $\boldsymbol{a}_\tau = \boldsymbol{0}$ between consecutive frames, where (1) can be rewritten as a linear function of $t$:

$$\boldsymbol{f}_{0\to t} = t\boldsymbol{f}_{0\to 1}. \tag{2}$$

However, the objects in real scenarios do not always travel in a straight line at a constant velocity,. Thus, these linear approaches cannot effectively model the complex non-uniform motion and often lead to inaccurate interpolation results.

In contrast, we take higher-order information into consideration and assume a constant $\boldsymbol{a}_\tau$ for $\tau \in [-1, 1]$. Correspondingly, the flow from frame 0 to $t$ can be derived as:

$$\boldsymbol{f}_{0\to t} = (\boldsymbol{f}_{0\to 1} + \boldsymbol{f}_{0\to -1})/2 \times t^2 + (\boldsymbol{f}_{0\to 1} - \boldsymbol{f}_{0\to -1})/2 \times t, \tag{3}$$

which is equivalent to temporally interpolating pixels with a quadratic function.

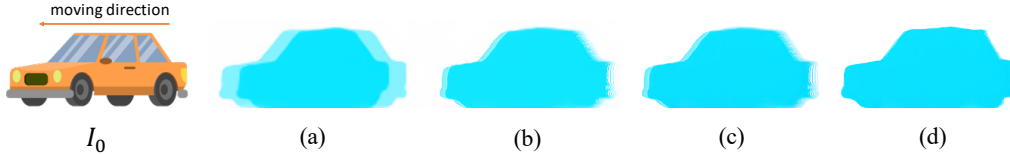

moving direction

$I_0$         (a)         (b)         (c)         (d)

Figure 3: Effectiveness of the flow reversal layer and the adaptive flow filtering. The car is moving along the arrow direction in the frame sequence. (a) is the $\boldsymbol{f}_{t\to 0}$ estimated with the naive strategy from [9]. (b) is the backward flow generated by our flow reversal layer. (c) and (d) represent the results of the deep CNNs in [9] and our adaptive flow filtering, respectively.

This formulation relaxes the constraint of constant velocity and rectilinear movement of linear models, and thus allows accelerated and curvilinear motion prediction between frames. In addition, existing methods with linear models only use the two closest frames $I_0, I_1$, whereas our algorithm naturally exploits visual information from more neighboring frames.

## 3.2   Flow reversal layer

While we obtain forward flow $\boldsymbol{f}_{0\to t}$ from quadratic flow prediction, it cannot be easily used for synthesizing images. Instead, we need backward flow $\boldsymbol{f}_{t\to 0}$ for high-quality frame synthesis [9, 14, 16, 31]. To estimate the backward flow, Jiang *et al.* [9] introduce a simple method which linearly combines $\boldsymbol{f}_{0\to 1}$ and $\boldsymbol{f}_{1\to 0}$ to approximate $\boldsymbol{f}_{t\to 0}$. However, this approach does not perform well around motion boundaries as shown in Figure 3(a). More importantly, this approach cannot be applied in our quadratic method to exploit the acceleration information.

In this work, we propose a flow reversal layer for better prediction of $\boldsymbol{f}_{t\to 0}$. We first project the flow map $\boldsymbol{f}_{0\to t}$ to frame $t$, where a pixel $\boldsymbol{x}$ on $I_0$ corresponds to $\boldsymbol{x} + \boldsymbol{f}_{0\to t}(\boldsymbol{x})$ on $I_t$. Next, we compute the flow of a pixel $\boldsymbol{u}$ on $I_t$ by reversing and averaging the projected flow values that fall into the neighborhood $\mathcal{N}(\boldsymbol{u})$ of pixel $\boldsymbol{u}$. Mathematically, this process can be written as:

$$\boldsymbol{f}_{t\to 0}(\boldsymbol{u}) = \frac{\sum_{\boldsymbol{x}+\boldsymbol{f}_{0\to t}(\boldsymbol{x})\in\mathcal{N}(\boldsymbol{u})} w(\|\boldsymbol{x}+\boldsymbol{f}_{0\to t}(\boldsymbol{x})-\boldsymbol{u}\|_2)(-\boldsymbol{f}_{0\to t}(\boldsymbol{x}))}{\sum_{\boldsymbol{x}+\boldsymbol{f}_{0\to t}(\boldsymbol{x})\in\mathcal{N}(\boldsymbol{u})} w(\|\boldsymbol{x}+\boldsymbol{f}_{0\to t}(\boldsymbol{x})-\boldsymbol{u}\|_2)}, \tag{4}$$

where $w(d) = e^{-d^2/\sigma^2}$ is the Gaussian weight for each flow. The proposed flow reversal layer is conceptually similar to the surface splatting [39] in computer graphics where the optical flow in our work is replaced by camera projection. During training, while the reversal layer itself does not have learnable parameters, it is differentiable and allows the gradients to be backpropagated to the flow estimation module in Figure 2, and thus enables end-to-end training of the whole system.

Note that the proposed reversal approach can lead to holes in the estimated flow map $\boldsymbol{f}_{t\to 0}$, which is mostly due to the objects visible in $I_t$ but occluded in $I_0$. And the missing objects are filled with the pixels warped from $I_1$ which is on the other side of the interpolation model.

## 3.3   Frame synthesis

In this section, we first refine the reversed flow field with adaptive filtering. Then we use the refined flow to generate the interpolated results with backward warping and frame fusion.

**Adaptive flow filtering.** While our approach is effective in reversing flow maps, the generated backward flow $\boldsymbol{f}_{t\to 0}$ may still have some ringing artifacts around edges as shown in Figure 3(b), which are mainly due to outliers in the original estimations of the PWC-Net. A straightforward way to reduce these artifacts [6, 9, 31] is to train deep CNNs with residual connections to refine the initial flow maps. However, this strategy does not work well in our practice as shown in Figure 3(c). This is because the artifacts from the flow reversal layer are mostly thin streaks with spike values (Figure 3(b)). Such outliers cannot be easily removed since the weighted averaging of convolution can be affected by the spiky outliers.

Inspired by the median filter [7] which samples only one pixel from a neighborhood and avoids the issues of weighted averaging, we propose a flow filtering network to adaptively sample the flow

map for removing outliers. While the classical median filter involves indifferentiable operation and cannot be easily trained in our end-to-end model, the proposed method learns to sample one pixel in a neighborhood with neural networks and can more effectively reduce the artifacts of the flow map.

Specifically, we formulate the adaptive filtering process as follows:

$$\boldsymbol{f}'_{t\to 0}(\boldsymbol{u}) = \boldsymbol{f}_{t\to 0}(\boldsymbol{u} + \boldsymbol{\delta}(\boldsymbol{u})) + \boldsymbol{r}(\boldsymbol{u}), \tag{5}$$

where $\boldsymbol{f}'_{t\to 0}$ denotes the filtered backward flow, and $\boldsymbol{\delta}(\boldsymbol{u})$ is the learned sampling offset of pixel $\boldsymbol{u}$. We constrain $\boldsymbol{\delta}(\boldsymbol{u}) \in [-k, k]$ by using $k \times \tanh(\cdot)$ as the activation function of $\boldsymbol{\delta}$ such that the proposed flow filter has a local receptive field of $2k + 1$. Since the flow map is sparse and smooth in most regions, we do not directly rectify the artifacts with CNNs as the schemes in [6, 9, 31]. Instead, we rely on the flow values around outliers by sampling in a neighborhood, where $\boldsymbol{\delta}$ is trained to find the suitable sampling locations. The residual map $\boldsymbol{r}$ is learned for further improvement. Our filtering method enables spatially-variant and nonlinear refinement of $\boldsymbol{f}_{t\to 0}$, which could be seen as a learnable median filter in spirit. As show in Figure 3(d), the proposed algorithm can effectively reduce the artifacts in the reversed flow maps. More implementation details are presented in Section 4.2.

**Warping and fusing source frames.** While we obtain $\boldsymbol{f}'_{t\to 0}$ with the input frames $I_{-1}$, $I_0$, and $I_1$, we can also estimate $\boldsymbol{f}'_{t\to 1}$ in a similar way with $I_0$, $I_1$, and $I_2$. Finally, we synthesize the intermediate video frames as:

$$\hat{I}_t(\boldsymbol{u}) = \frac{(1 - t)m(\boldsymbol{u})I_0(\boldsymbol{u} + \boldsymbol{f}'_{t\to 0}(\boldsymbol{u})) + t(1 - m(\boldsymbol{u}))I_1(\boldsymbol{u} + \boldsymbol{f}'_{t\to 1}(\boldsymbol{u}))}{(1 - t)m(\boldsymbol{u}) + t(1 - m(\boldsymbol{u}))} \tag{6}$$

where $I_i(\boldsymbol{u} + \boldsymbol{f}'_{t\to i}(\boldsymbol{u}))$ denotes the pixel warped from frame $i$ to $t$ with bilinear function [8]. $m$ is a mask learned with a CNN to fuse the warped frames. Similar to [9], we also use the temporal distance $1 - t$ and $t$ for the source frames $I_0$ and $I_1$, such that we can give higher confidence to temporally-closer pixels. Note that we do not directly use the pixels in $I_{-1}$ and $I_2$ for image synthesis, as almost all the contents in the intermediate frame can be found in $I_0$ and $I_1$. Instead, $I_{-1}$ and $I_2$ are exploited for acceleration-aware motion estimation.

Since all the above steps of our method are differentiable, we can train the proposed interpolation model in an end-to-end manner. The loss function for training our network is a combination of the $\ell_1$ loss and perceptual loss [10, 37]:

$$\|\hat{I}_t - I_t\|_1 + \lambda \|\phi(\hat{I}_t) - \phi(I_t)\|_2, \tag{7}$$

where $I_t$ is the ground truth, and $\phi$ is the `conv4_3` feature extractor of the VGG16 model [28].

## 4    Experiments

In this section, we first provide implementation details of the proposed model, including training data, network structure, and hyper-parameters. We then present evaluation results of our algorithm with comparisons to the state-of-the-art methods on video datasets. The source code, data, and the trained models are available at: https://sites.google.com/view/xiangyuxu/qvi_nips19.

### 4.1    Training data

To train the proposed interpolation model, we collect high-quality videos from the Internet, where each frame is of 1080×1920 pixels at the frame rate of 960 fps. From the collected videos, we select the clips with both camera shake and dynamic object motion, which are beneficial for more effective network training. The final training dataset consists of 173 video clips of different scenes and 36926 frames in total. In addition, the 960 fps video clips are randomly downsampled to 240 fps and 480 fps for data augmentation. During the training process, we extract non-overlapped frame groups from these video clips, where each has 4 input frames $I_{-1}$, $I_0$, $I_1$, $I_2$, and 7 target frames $I_t$, $t = 0.125, 0.25, \ldots, 0.875$. We resize the frames into 360×640 and randomly crop 352×352 patches for training. Image flipping and sequence reversal are also performed to fully utilize the video data.

### 4.2    Implementation details

We learn the adaptive flow filtering with a 23-layer U-Net [26, 36] which is an encoder-decoder network. The encoder is composed of 12 convolution layers with 5 average pooling layers for

Table 1: Quantitative evaluations on the GOPRO and Adobe240 datasets. "Ours w/o qua." represents our model without using the quadratic flow prediction.

| Method | GOPRO | | | | | | Adobe240 | | | | | |
| | whole | | | center | | | whole | | | center | | |
| | PSNR | SSIM | IE | PSNR | SSIM | IE | PSNR | SSIM | IE | PSNR | SSIM | IE |
|---|---|---|---|---|---|---|---|---|---|---|---|---|
| Phase | 23.95 | 0.700 | 17.89 | 22.05 | 0.620 | 22.08 | 25.60 | 0.735 | 16.93 | 23.65 | 0.647 | 20.65 |
| DVF | 21.94 | 0.776 | 21.30 | 20.55 | 0.720 | 25.14 | 28.23 | 0.896 | 11.76 | 26.90 | 0.871 | 13.30 |
| SepConv | 29.52 | 0.922 | 9.26 | 27.69 | 0.895 | 11.38 | 32.19 | 0.954 | 7.71 | 30.87 | 0.940 | 8.91 |
| SuperSloMo | 29.00 | 0.918 | 9.51 | 27.33 | 0.892 | 11.50 | 31.30 | 0.949 | 8.18 | 30.17 | 0.935 | 9.22 |
| Ours w/o qua. | 29.57 | 0.923 | 9.02 | 27.86 | 0.898 | 10.93 | 31.64 | 0.952 | 7.93 | 30.48 | 0.939 | 8.96 |
| Ours | **31.27** | **0.948** | **7.23** | **29.62** | **0.929** | **8.73** | **32.95** | **0.966** | **6.84** | **32.09** | **0.959** | **7.47** |

dowsampling, and the decoder has 11 convolution layers with 5 bilinear layers for upsampling. We add skip connections with pixel-wise summation between the same-resolution layers in the encoder and decoder to jointly use low-level and high-level features. The input of our network is a concatenation of $I_0$, $I_1$, $I_0^t$, $I_1^t$, $\boldsymbol{f}_{0\to1}$, $\boldsymbol{f}_{1\to0}$, $\boldsymbol{f}_{t\to0}$, and $\boldsymbol{f}_{t\to1}$, where $I_i^t(\boldsymbol{u}) = I_i(\boldsymbol{u} + \boldsymbol{f}_{t\to i}(\boldsymbol{u}))$ denotes the pixel warped with $\boldsymbol{f}_{t\to i}$. The U-Net produces the output $\boldsymbol{\delta}$ and $\boldsymbol{r}$ which are used to estimate the filtered flow map $\boldsymbol{f}'_{t\to0}$ and $\boldsymbol{f}'_{t\to1}$ with (5). Then we warp $I_0$ and $I_1$ with flow $\boldsymbol{f}'_{t\to0}$ and $\boldsymbol{f}'_{t\to1}$, and feed the warped images to a 3-layer CNN to estimate the fusion mask $m$ which is finally used for frame interpolation with (6).

We first train the proposed network with the flow estimation module fixed for 200 epochs, and then finetune the whole system for another 40 epochs. Similar to [34], we use the Adam optimizer [12] for training. We initialize the learning rate as $10^{-4}$ and further decrease it by a factor of $0.1$ at the end of the $100^{th}$ and $150^{th}$ epochs. The trade-off parameter $\lambda$ of the loss function (7) is set to be $0.005$. $k$ in the activation function of $\boldsymbol{\delta}$ is set to be $10$. In the flow reversal layer, we set the Gaussian standard deviation $\sigma = 1$.

For evaluation, we report PSNR, Structural Similarity Index (SSIM) [32], and the interpolation error (IE) [2] between the predictions and ground truth intermediate frames, where IE is defined as root-mean-squared (RMS) difference between the reference and interpolated image.

## 4.3 Comparison with the state-of-the-arts

We evaluate our model with the state-of-the-art video interpolation approaches, including the phase-based method (Phase) [17], separable adaptive convolution (SepConv) [21], deep voxel flow (DVF) [14], and SuperSloMo [9]. We use the original implementations for Phase, SepConv, DVF and the implementation from [22] for SuperSlomo. We retrain DVF and SuperSlomo with our data. We were not able to retrain SepConv as the training code is not publicly available, and directly use the original models [21] in our experiments instead. Note that the proposed quadratic video interpolation can be used for synthesizing arbitrary intermediate frames, which is evaluated on the high frame rate video datasets such as GOPRO [18] and Adobe240 [30]. We also conduct experiments on the UCF101 [29] and DAVIS [23] datasets for performance evaluation of single-frame interpolation.

**Multi-frame interpolation on the GOPRO [18] dataset.**    This dataset is composed of 33 high-quality videos with a frame rate of $720\,\mathrm{fps}$ and image resolution of $720\times1280$. These videos are recorded with hand-held cameras, which often contain non-linear camera motion. In addition, this dataset has dynamic object motion from both indoor and outdoor scenes, which are challenging for existing interpolation algorithms.

We extract 4275 non-overlapped frame sequences with a length of 25 from the GOPRO videos. To evaluate the proposed quadratic model, we use the $1^{st}$, $9^{th}$, $17^{th}$, and $25^{th}$ frames of each sequence as our inputs, which respectively correspond to $I_{-1}, I_0, I_1, I_2$ in the proposed model. As discussed in Section 1, the baseline methods only exploit the $9^{th}$ and $17^{th}$ frames for video interpolation. We synthesize 7 frames between the $9^{th}$ and $17^{th}$ frames, and thus all the corresponding ground truth frames are available for evaluation.

As shown in Table 1, we separately evaluate the scores of the center frame (*i.e.,* the $4^{th}$ frame, denoted as center) and the average of all the 7 interpolated frames (denoted as whole). The quadratic interpolation model consistently performs favorably against all the other linear methods. Noticeably,

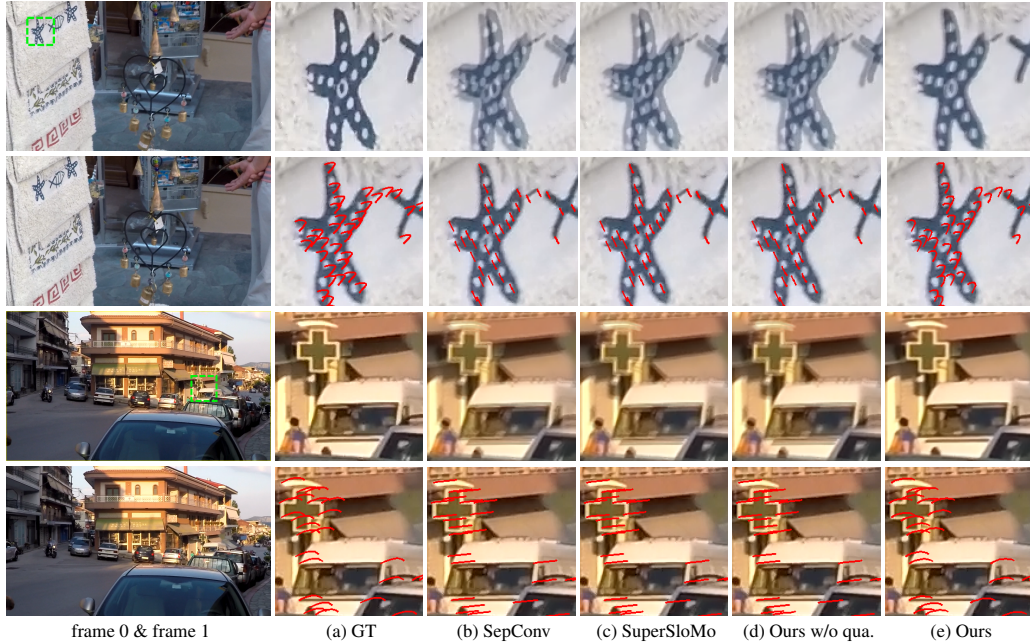

| frame 0 & frame 1 | (a) GT | (b) SepConv | (c) SuperSloMo | (d) Ours w/o qua. | (e) Ours |

Figure 4: Qualitative results on the GOPRO dataset. The first row of each example shows the overlap of the interpolated center frame and the ground truth. A clearer overlapped image indicates more accurate interpolation result. The second row of each example shows the interpolation trajectory of all the 7 interpolated frames by feature point tracking.

the PSNRs of our results on either the center frame or the average of the whole frames improve over the second best method by more than $1\,\mathrm{dB}$.

To understand the effectiveness of the proposed quadratic interpolation algorithm, we visualize the trajectories of the interpolated results in Figure 4 and compare with the baseline methods both qualitatively and quantitatively. Specifically, for each test sequence from the GOPRO dataset, we use the classic feature tracking algorithm [27] to select 10000 feature points in the 9th frame, and track them through the 7 synthesized in-between frames. For better performance evaluation, we exclude the points that disappear or move out of the image boundaries during tracking.

We show two typical examples in Figure 4 and visualize the interpolation trajectory by connecting the tracking points (*i.e.,* the red lines). In the the first example, the object moves along a quite sharp curve, mostly due to a sudden violent change of the camera's moving direction. All the existing methods fail on this example as the linear models assume uniform motion and cannot predict the motion change well. In contrast, our quadratic model enables higher-order video interpolation and exploits the acceleration information from the neighboring frames. As shown in Figure 4(e), the proposed method approximates the curvilinear motion well against the ground truth.

In addition, we overlap the predicted center frame with its ground truth to evaluate the interpolation accuracy of different methods (first row of each example of Figure 4). For linear models, the overlapped frames are severely blurred, which demonstrates the large shift between ground truth and the linearly interpolated results. In contrast, the generated frames by our approach align with the ground truth well, which indicates better interpolation results with smaller errors.

Different from the first example which contains severe non-linear movements, we present a video with motion trajectory closer to straight lines in the second example. As shown in Figure 4, although the motion in this video is closer to the uniform assumption of linear models, existing approaches still do not generate accurate interpolation results. This demonstrates the importance of the proposed quadratic algorithm, since there are few scenes strictly satisfying uniform motion, and minor per-turbations to this strict motion assumption can lead to obvious shifts in the synthesized images. As shown in the second example of Figure 4(e), the proposed quadratic method estimates the moving trajectory well against the ground truth and thus generates more accurate interpolation results.

Table 2: ASFP on the GOPRO dataset.

| Method | whole | center |
|---|---|---|
| SepConv | 1.79 | 2.17 |
| SuperSloMo | 2.04 | 2.38 |
| Ours w/o qua. | 1.33 | 1.69 |
| Ours | **0.97** | **1.22** |

Table 3: Evaluations on the UCF101 and DAVIS datasets.

| Method | UCF101 | | | DAVIS | | |
|---|---|---|---|---|---|---|
| | PSNR | SSIM | IE | PSNR | SSIM | IE |
| Phase | 29.84 | 0.900 | 7.97 | 21.54 | 0.556 | 26.76 |
| DVF | 29.88 | 0.916 | 7.66 | 22.24 | 0.742 | 23.66 |
| SepConv | 31.97 | 0.943 | 5.89 | 26.21 | 0.857 | 15.84 |
| SuperSloMo | 32.04 | 0.945 | 5.99 | 25.76 | 0.850 | 15.93 |
| Ours w/o qua. | 32.02 | 0.945 | 5.99 | 26.83 | 0.874 | 13.69 |
| Ours | **32.54** | **0.948** | **5.79** | **27.73** | **0.894** | **12.32** |

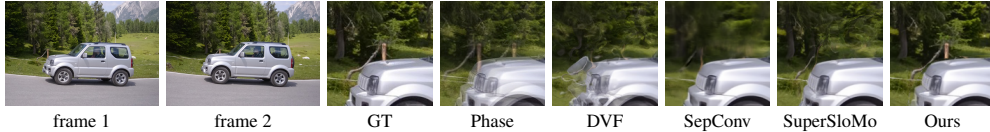

frame 1     frame 2     GT     Phase     DVF     SepConv     SuperSloMo     Ours

Figure 5: Visual results from the DAVIS dataset.

In addition, if we do not consider the acceleration in the proposed method (*i.e.,* using (2) to replace (3)), the interpolation performance of our model decreases drastically to that of linear models ("Ours w/o qua." in Table 1 and Figure 4), which shows the importance of the higher-order information.

To quantitatively measure the shifts between the synthesized frames and ground truth, we define a new error metric for video interpolation denoted as average shift of feature points (ASFP):

$$ASFP(I_t, \hat{I}_t) = \frac{1}{N} \sum_{i=1}^{N} \|\boldsymbol{p}(I_t, i) - \boldsymbol{p}(\hat{I}_t, i)\|_2, \tag{8}$$

where $\boldsymbol{p}(I_t, i)$ denotes the position of the $i^{th}$ feature point on $I_t$, and $N$ is the number of feature points. We respectively compute the average ASFP of the center frame and the whole 7 interpolated frames on the GOPRO dataset. Table 2 shows the proposed quadratic algorithm performs favorably against the state-of-the-art methods while significantly reducing the average shift.

**Evaluations on the Adobe240 [30] dataset.** This dataset consists of 133 videos with a frame rate of 240 fps and image resolution of 720×1280 pixels. The frames are resized to 360×480 during testing. We extract 8702 non-overlapped frame sequences from the videos in the Adobe240 dataset, and each sequence contain 25 consecutive frames similar with the settings of the GOPRO dataset. We also synthesize 7 in-between frames for 8 times temporally upsampling. As shown in Table 1, the proposed quadratic algorithm performs favorably against the state-of-the-art linear interpolation methods.

**Single-frame interpolation on the UCF101 [29] and DAVIS [23] datasets.** In addition to the multi-frame interpolation evaluated on high frame rate videos, we test the proposed quadratic model on single-frame interpolation using videos with 30 fps, *i.e.,* UCF101 [29] and DAVIS [23] datasets.

Liu *et al.* [14] previously extract 100 triplets from the videos of the UCF101 dataset as test data, which cannot be used to evaluate our algorithms since we need four consecutive frames as inputs. Thus, we re-generate the test data by first temporally downsampling the original videos to 15 fps and then randomly extracting 4 adjacent frames (*i.e.,* $I_{-1}, I_0, I_1, I_2$) from these videos. The sequences with static scenes are removed for more accurate evaluations. We collect 100 quintuples (4 input frames $I_{-1}, I_0, I_1, I_2$ and 1 target frame $I_{0.5}$) where each frame is resized to 225×225 pixels as [14]. For the DAVIS dataset, we evaluate our method on the whole 90 video clips which are divided into 2847 quintuples using the original image resolution.

We interpolate the center frame for these two dataset, which is equivalent to converting a 15 fps video to a 30 fps one. As shown in Table 3, the quadratic interpolation approach performs slightly better than the baseline models on the UCF101 dataset as the videos are of relatively low quality with low image resolution and slow motion. For the DAVIS dataset which contains complex motion from both camera shake and dynamic scenes, our method significantly outperforms other approaches in terms of all evaluation metrics. We show one example from the DAVIS dataset in Figure 5 for visual comparisons.

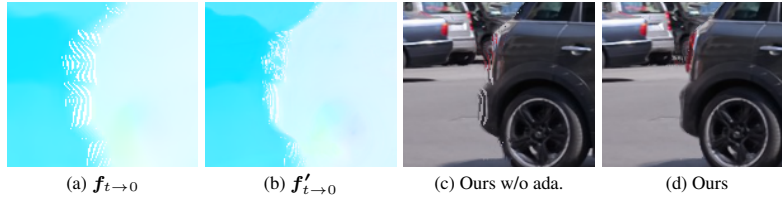

| (a) $\boldsymbol{f}_{t\to 0}$ | (b) $\boldsymbol{f}'_{t\to 0}$ | (c) Ours w/o ada. | (d) Ours |

Figure 6: Adaptive flow filtering reduces artifacts in (a) and generates higher-quality image (d).

Overall, the quadratic approach achieves state-of-the-art performance on a wide variety of video datasets for both single-frame and multi-frame interpolations. More importantly, experimental results demonstrate that it is important and effective to exploit the acceleration information for accurate video frame interpolation.

## 4.4 Ablation study

We analyze the contribution of each component in our model on the DAVIS video dataset [23] in Table 4. In particular, we study the impact of quadratic interpolation by replacing the quadratic flow prediction (3) with the linear function (2) (w/o qua.). We further study the effectiveness of the adaptive flow filtering by directly learning residuals for flow refinement similar with [6, 9, 31] (w/o ada.). In addition, we compare the flow reversal layer with the linear

Table 4: Ablation study on the DAVIS dataset.

| Method | PSNR | SSIM | IE |
|---|---|---|---|
| Ours w/o rev. | 26.71 | 0.873 | 13.84 |
| Ours w/o qua. | 26.83 | 0.874 | 13.69 |
| Ours w/o ada. | 27.60 | 0.892 | 12.41 |
| Ours full model | **27.73** | **0.894** | **12.32** |

combination strategy in [9] which approximates $\boldsymbol{f}_{t\to 0}$ by simply fusing $\boldsymbol{f}_{0\to 1}$ and $\boldsymbol{f}_{1\to 0}$ (w/o rev.). As shown in Table 4, removing each of the three components degrades performance in all metrics. Particularly, the quadratic flow prediction plays a crucial role, which verifies our approach to exploit the acceleration information from additional neighboring frames. Note that while the quantitative improvement from the adaptive flow filtering is small, this component is effective in generating high-quality interpolation results by reducing artifacts of the flow fields as shown in Figure 6.

## 5 Conclusion

In this paper we propose a quadratic video interpolation algorithm which can synthesize high-quality intermediate frames. This method exploits the acceleration information from neighboring frames of a video for non-linear video frame interpolation, and facilitates end-to-end training. The proposed method is able to model complex motion in real world more accurately and generate more favorable results than existing linear models on different video datasets. While we focus on quadratic function in this work, the proposed formulation is general and can be extended to even higher-order interpolation methods, *e.g.,* the cubic model. We also expect this framework to be applied to other related tasks, such as multi-frame optical flow and novel view synthesis.

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
