[Reviews · NeurIPS 2019]

Reviewer 1



This work proposes a method of estimating and using the higher-order information, i.e. acceleration, for optical flow estimation such that the interpolated frames can capture motions more naturally. The idea is interesting and straightforward and I am surprised that no one has done this before. The work is very well presented with sufficient experiments. The SM is well prepared. The flow reversal layer is somehow novel, but it is not very clear what exactly learned by the reversal layer. What is the performance of the learned layer compared to a reversal layer with fixed parameters? I am not sure about the contribution of the adaptive flow filtering. It is hard to get the logic why the proposed method is a better way of reducing artifacts. The result in Table 4 also shows very marginal improvements by using this adaptive flow filtering. It will be great to see the same ablation study on other datasets.

Reviewer 2



Positive: - Clear and important motivation and problem. - Good writing throughout; concise, to the point. Thank you. - Method is simple. - Improvement from integration of multi-frame linear flow vs. quadratic flow is tested - Clear results on multiple datasets, with ablated components. - Video: Results from 1m 50s onwards are great. This presentation format with the comparison to other techniques would be good to use for the boat sequence, also. Room for improvement: - Method does not directly estimate quadratic flow, but instead estimates linear flow across pairs of frames and combines them. - SepConv and DVF were not retrained on your 'large motion' clip database. It is known that better training data (more selective to the problem) is important. What effect does this 'large motion' clip selection have on these models? Put another way, what is the difference between the SuperSloMo network trained on regular vs. large motion data? - Video results: 42 seconds: while impressive, this result shows visible popping/jumping whenever we reach an original video frame. Why? - Video results: The interpolated frames are noticeably blurrier than the original frames. While this is a problem with these techniques in general (potentially caused by the averaging model for frame synthesis), it is still a limitation and should be stated as such. Minor: - Title: consider adding 'temporal' or 'motion' to the title. - L86: additional comma at the end of the line. - Figure 4 - the frames labeled as GT with the trajectories: the trajectories are estimated and should be labeled as such; there are no GT trajectories. - L225 - please add Eq. (2) not just (2)

Reviewer 3



+ Novel idea + Good results + complete evaluation - Technical section can be streamlined more - flow filtering missing some technical details I liked the idea and motivation of the paper. It makes intuitive sense and is well explained. The proposed algorithm produces good results, not just numerically, but also qualitatively. There is an extensive comparison to prior work and a ablation study. One minor request would be to ablate all possible combinations of components instead of just removing each individual component. It would make the paper stronger. However, the current ablation is sufficient. May main points of improvement lie in the technical section. First, equation 1 is probably not needed. Most readers will be familiar with velocity and acceleration. Maybe, a slightly version of equation 3 is fine. If the authors decide to keep equation 1, I'd recommend to fix the notation: the bounds of the integration currently use the same symbol as the differential, which seems wrong. The proposed flow reversal (eq 4) is commonly called splatting in computer graphics. It might be worth a citation, and carefully highlighting the differences (if there are any). If it is just splatting of the flow field (which seems to be the case at first glance), the section can be shortened a bit. Finally, I was missing some information on how the flow filtering works, how the network is set up, what are the inputs, what are the outputs. Are all inputs fed into a single network (concatenated or separate paths), is there any normalization between images and flow fields?

[Author Response · NeurIPS 2019]

We thank all reviewers for their constructive comments and address the raised issues below.

**R1: More descriptions of the flow reversal layer.** As described in Secion 3.2 of the manuscript, we introduce the flow reversal layer to synthesize intermediate video frames. During training, while the reversal layer itself does not have learnable parameters, it allows the gradients to be backpropagated to the flow estimation module in Figure 2 of the manuscript, and thus enables end-to-end training of the whole system. The finetuned flow estimation network improves the PSNR by $0.15$ and $0.17 \, \mathrm{dB}$ on the "whole" and "center" of the Adobe240 dataset against the model with parameters fixed. More importantly, the flow reversal layer can estimate the backward flow from the acceleration-aware forward flow, which is in sharp contrast to the linear combination strategy of [6]. This significantly improves the results as shown in Table 4 of the manuscript. The source code, as mentioned on L141, will be made available to the public.

**R1: Why the adaptive flow filtering is a better way of reducing artifacts?** As introduced on L112-120 of our paper, the artifacts from the flow reversal layer are mostly thin streaks with spike values (Figure 1(a)). Such outliers cannot be easily removed by convolution layers (Figure 1(b)) because the weighted averaging of convolution can be affected by the spike outliers. In image processing, the outliers with spike values (*e.g.* salt-and-pepper noise) are usually handled by median filters which sample one pixel from a neighborhood and avoid the issues of weighted averaging (Gonzalez *et al.*, Digital Image Processing, 2002). However, the median filter involves indifferentiable operation, and cannot be easily trained in our end-to-end model. In contrast, the proposed adaptive flow filtering samples one pixel in a neighborhood by learning the sampling location with neural networks and can more effectively reduce the artifacts of the flow map (Figure 1(c)). Our method could be seen as a learnable median filter in spirit.

(a) $\boldsymbol{f}_{t \to 0}$     (b) $\boldsymbol{f}'_{t \to 0}$ with conv.     (c) $\boldsymbol{f}'_{t \to 0}$ with ada.     (d) Result with (b)     (e) Result with (c)

Figure 1: Effectiveness of the adaptive flow filtering. (a) is the backward flow $\boldsymbol{f}_{t \to 0}$. (b) is the filtered flow field by a CNN with residual connection. (c) is produced by the proposed adaptive flow filtering. (d) and (e) are the synthesized results with (b) and (c), respectively.

**R1: The same ablation study on other datasets.** We present an ablation study on the Adobe240 in Table 1, which shows similar results to Table 4 of the manuscript. Although the quantitative improvement from the adaptive flow filtering (ada.) is small, this component is important in generating results with higher visual quality (Figure 1(d) and (e)).

Table 1: Ablation study on the Adobe240 dataset.

| Method | whole | | | center | | |
|---|---|---|---|---|---|---|
| | PSNR | SSIM | IE | PSNR | SSIM | IE |
| Ours w/o rev. | 31.32 | 0.950 | 8.12 | 30.08 | 0.936 | 9.23 |
| Ours w/o qua. | 31.28 | 0.950 | 8.18 | 30.16 | 0.937 | 9.21 |
| Ours w/o ada. | 32.72 | **0.965** | 6.94 | 31.89 | **0.958** | 7.57 |
| Ours | **32.81** | **0.965** | **6.90** | **31.96** | **0.958** | **7.52** |

**R2: Directly estimate quadratic flow instead of combining linear flow.** As suggested, we will explore this interesting idea in the future work.

**R2: SepConv and DVF were not retrained.** While the DVF was originally trained on a low-quality dataset UCF-101, SepConv has originally been trained on high-quality videos with large motion. As suggested, we retrain DVF on the proposed dataset. The PSNRs of the retrained DVF are $28.05$ and $26.81 \, \mathrm{dB}$ on "whole" and "center" of the Adobe240 dataset. We were not able to retrain SepConv in the rebuttal phase as only the test code is released and the training code is not publicly available. We will implement SepConv by ourselves and update the results in the revised paper.

**R2: The output frames are blurrier than the original frames.** This issue may be caused by the averaging model for frame synthesis which has been used in most video interpolation models. One possible solution to remedy the problem is to add a GAN loss to encourage sharper results. We will discuss the limitations in the revised paper.

**R3: Relation to splatting.** The proposed flow reversal layer is conceptually similar to the surface splatting in computer graphics where the optical flow in our work is replaced by camera projection. We will add the corresponding reference in the revised paper.

**R3: More details of the flow filtering network.** The flow filtering network is a 23-layer U-Net, where the encoder is composed of 12 convolution layers and 5 average pooling layers for dowsampling, and the decoder has 11 convolution layers as well as 5 bilinear layers for upsampling. The input of our network is a concatenation of $I_0$, $I_1$, $I'_0$, $I'_1$, $\boldsymbol{f}_{0 \to 1}$, $\boldsymbol{f}_{1 \to 0}$, $\boldsymbol{f}_{t \to 0}$, and $\boldsymbol{f}_{t \to 1}$, where $I'_0$ and $I'_1$ are the warped $I_0$ and $I_1$ with flow $\boldsymbol{f}_{t \to 0}$ and $\boldsymbol{f}_{t \to 1}$. We do not apply any normalization between images and flow fields. The U-Net produces the output $\boldsymbol{\delta}$ and $\boldsymbol{r}$ which are used to estimate the filtered flow map $\boldsymbol{f}'_{t \to 0}$ and $\boldsymbol{f}'_{t \to 0}$ with Eq. 5. Then we warp $I_0$ and $I_1$ with flow $\boldsymbol{f}'_{t \to 0}$ and $\boldsymbol{f}'_{t \to 1}$, and feed the warped images to a 3-layer CNN to estimate the fusion mask $m$ which is finally used for frame interpolation with Eq. 6. More detailed descriptions will be added in the revised paper.

[Meta-Review · NeurIPS 2019]

Reviews agree that this paper provides a technically elegant improvement to the state-of-the-art in video interpolation, an important problem with a wide range of practical applications. Results are qualitatively extremely impressive, in the supplemental video the video interpolation quality improvements over previous work is dramatic; this result should be of wide interest. Among its technical innovations, the paper introduces a novel "flow reversal" layer needed for video interpolation, and show how this differentiable mapping can be integrated in the deep motion prediction network.